🔓 | **Open Peer Review** | Antimicrobial Chemotherapy | Research Article

# HOCl-producing electrochemical bandage is active in murine polymicrobial wound infection

Derek Fleming,[1] Ibrahim Bozyel,[2] Christina A. Koscianski,[1] Dilara Ozdemir,[2] Melissa J. Karau,[1] Luz Cuello,[1] Md Monzurul Islam Anoy,[2] Suzanne Gelston,[2] Audrey N. Schuetz,[1] Kerryl E. Greenwood-Quaintance,[1] Jayawant N. Mandrekar,[3] Haluk Beyenal,[2] Robin Patel[1,4]

**ABSTRACT** Wound infections, exacerbated by the prevalence of antibiotic-resistant bacterial pathogens, necessitate innovative antimicrobial approaches. Polymicrobial infections, often involving *Pseudomonas aeruginosa* and methicillin-resistant *Staphylococcus aureus* (MRSA), present challenges due to biofilm formation and antibiotic resistance. Hypochlorous acid (HOCl), a potent antimicrobial agent, holds promise as an alternative therapy. An electrochemical bandage (e-bandage) that generates HOCl *in situ* via precise polarization controlled by a miniaturized potentiostat was evaluated for the treatment of murine wound biofilm infections containing both *P. aeruginosa* with "difficult-to-treat" resistance and MRSA. Previously, HOCl-producing e-bandage was shown to reduce murine wound biofilms containing *P. aeruginosa* alone. Here, in 5-mm excisional skin wounds containing 48-h biofilms comprising MRSA and *P. aeruginosa* combined, polarized e-bandage treatment reduced MRSA by 1.1 $\log_{10}$ CFU/g ($P = 0.026$) vs non-polarized e-bandage treatment (no HOCl production), and 1.4 $\log_{10}$ CFU/g (0.0015) vs Tegaderm only controls; *P. aeruginosa* was similarly reduced by 1.6 $\log_{10}$ CFU/g ($P = 0.0032$) and 1.6 $\log_{10}$ CFU/g ($P = 0.0015$), respectively. For wounds infected with MRSA alone, polarized e-bandage treatment reduced bacterial load by 1.1 $\log_{10}$ CFU/g ($P = 0.0048$) and 1.3 $\log_{10}$ CFU/g ($P = 0.0048$) compared with non-polarized e-bandage and Tegaderm only, respectively. The e-bandage treatment did not negatively impact wound healing or cause tissue toxicity. The addition of systemic antibiotics did not enhance the antimicrobial efficacy of e-bandages. This study provides additional evidence for the HOCl-producing e-bandage as a novel antimicrobial strategy for managing wound infections, including in the context of antibiotic resistance and polymicrobial infections.

**IMPORTANCE** New approaches are needed to combat the rise of antimicrobial-resistant infections. The HOCl-producing electrochemical bandage (e-bandage) leverages *in situ* generation of HOCl, a natural biocide, for broad-spectrum killing of wound pathogens. Unlike traditional therapies that may exhibit limited activity against biofilms and antimicrobial-resistant organisms, the e-bandage offers a potent, standalone solution that does not contribute to further resistance or require adjunctive antibiotic therapy. Here, we show the ability of the e-bandage to address polymicrobial infection by antimicrobial resistant clinical isolates of *Staphylococcus aureus* and *Pseudomonas aeruginosa*, two commonly isolated, co-infecting wound pathogens. Effectiveness of the HOCl-producing e-bandage in reducing pathogen load while minimizing tissue toxicity and avoiding the need for systemic antibiotics underscores its potential as a tool in managing complex wound infections.

**KEYWORDS** electrochemical bandage, hypochlorous acid, *Pseudomonas aeruginosa*, methicillin-resistant *Staphylococcus aureus*, anti-biofilm, *in vivo* wound infection

Address correspondence to Robin Patel, patel.robin@mayo.edu.

R.P. reports grants from MicuRx Pharmaceuticals and BIOFIRE. R.P. is a consultant to PhAST, Day Zero Diagnostics, Abbott Laboratories, Sysmex, DEEPULL DIAGNOSTICS, S.L., Netflix, HealthTrackRx, and CARB-X. In addition, R.P. has a patent for *Bordetella pertussis/parapertussis* PCR issued, a patent for a device/method for sonication with royalties paid by Samsung to Mayo Clinic, and a patent on an anti-biofilm substance issued. R.P. receives honoraria from Up-to-Date and the Infectious Diseases Board Review Course. H.B. holds a patent, "Electrochemical reduction or prevention of infections" (US20180207301A1), which refers to the electrochemical scaffold upon which the current design of e-bandage is based.

See the funding table on p. 14.

The emergence of bacteria that are resistant to antibiotics demands new antimicrobial strategies. This is particularly critical in the context of wound infections. Almost 90% of wound samples may carry microorganisms with resistance to at least one antibiotic, with about 30% exhibiting resistance to six or more antibiotics (1). Among these, the highly prevalent Gram-positive pathogen, methicillin-resistant *Staphylococcus aureus* (MRSA), and *Pseudomonas aeruginosa*, a Gram-negative pathogen that is intrinsically resistant to multiple antibiotics and prone to acquiring resistance (2), are frequently identified in wound infections (3, 4).

The presence of biofilms, communities of microorganisms protected by a complex matrix of extracellular polysaccharides, proteins, DNA, and other substances called extracellular polymeric substance (EPS), further potentiates resistance in pathogens, such as *P. aeruginosa* and MRSA. Biofilm-related infections can be challenging to treat with existing therapies, hindering wound healing and causing persistent inflammation (5, 6). In the United States, ~7 million people suffer from chronic wounds annually, with a majority of these wounds associated with biofilms (7, 8). Given the recalcitrance of chronic wound infections, and the common involvement of antibiotic-resistant *P. aeruginosa* and MRSA, it is essential to develop new antibiofilm strategies that do not contribute to further antibiotic resistance.

Hypochlorous acid (HOCl) is a reactive oxygen species (ROS), naturally produced by phagocytes, that has potent antimicrobial properties (9, 10). It has been shown to be broadly effective at killing bacterial and fungal pathogens (11–13). A barrier to clinical use has been the inability to continuously deliver microbicidal, non-toxic concentrations to infection sites. In past studies, we developed an electrochemical bandage (e-bandage) that generates HOCl *in situ*, and showed it to be active against bacterial and fungal biofilms *in vitro*, and against *in vivo* murine excisional skin wounds infected with *P. aeruginosa* alone (11–16). Wound infections are commonly polymicrobial, increasing recalcitrance and complicating treatment (1, 17). Here, the HOCl-producing e-bandage, controlled by a miniature, "micropotentiostat," was effective in treating murine excisional wound biofilm infections containing *P. aeruginosa* and MRSA together (and MRSA alone). *S. aureus* and *P. aeruginosa* are the two most commonly co-isolated wound pathogens (1, 4, 17, 18). Assessment involved measuring amounts of live bacteria within wounds, examining the progress of wound healing through reduction of wound size, scoring of purulence reduction, analyzing tissue histopathology, and measuring levels of blood biochemistry markers and inflammatory cytokines. Additionally, the concentration of HOCl in wounds was measured, and scanning electron microscopy (SEM) was conducted on excised wound biofilms to evaluate treatment impact on the biofilm matrix and integrity, and abundance of bacterial cells. Finally, HOCl-producing e-bandage treatment was compared with systemic antibiotic treatment, and the ability of e-bandage treatment to potentiate concurrently administered systemic antibiotics evaluated.

## MATERIALS AND METHODS

### Electrochemical bandage

The e-bandage and micropotentiostat have been previously described (14–16). Briefly, the e-bandage comprises carbon fabric electrodes (Panex 30 PW-06, Zoltek Companies Inc., St. Louis, MO), with surfaces measuring 1.77 cm$^2$ each for the working electrode (WE) and counter electrode (CE), along with a silver/silver chloride (Ag/AgCl) wire serving as a quasi-reference electrode (RE). Subsequently, 400-µL hydrogel (saline + 1.8% xanthan gum) is injected on top of the wound bed and between the WE and CE. A mciropotentiostat, powered by a 3-V coin cell battery, maintains the operational potential of the working electrode at +1.5 $V_{Ag/AgCl}$. The carbon fabric electrodes are separated by two layers of cotton fabric, with an additional layer placed over the CE to aid in moisture retention. These layers are secured using a silicone adhesive. The RE is positioned between the cotton fabric layers separating the carbon electrodes. Titanium wires (TEMCo, Amazon.com, catalog #RW0524) with nylon sew-on caps (Dritz,

Spartanburg, SC, item#85) connect to opposite ends of the e-bandage and link to the potentiostat (Fig. 1). Under physiological conditions, polarization of the bandage leads to the generation of HOCl through these reactions:

$$2Cl^- \Leftrightarrow Cl_2 + 2e^- \qquad E_0 = -1.16\ V_{Ag/AgCl}$$

$$Cl_2 + H_2O \Leftrightarrow Cl^- + HOCl + H^+$$

At pH 7.4 and 25°C, the conditions under which e-bandage was utilized, HOCl dissociates to ~57% HOCl and ~43% ClO⁻ (19).

## Mouse skin wound infection model

All animal experiments were approved by the Mayo Clinic Institutional Animal Care and Use Committee (A00003272-20). Full-thickness skin wounds were generated on Swiss Webster mice (Charles River, Wilmington, MA). The animals were anesthetized by intraperitoneal injection of a mixture of ketamine (90 mg/kg) and xylazine (10 mg/kg). Subcutaneous buprenorphine ER-Lab (1 mg/kg) was administered for analgesia. Mature wound biofilms were created as previously reported (20, 21). The dorsal surface was shaved and disinfected, and a circular full-thickness skin wound created using a 5-mm biopsy punch (Acuderm Inc., Fort Lauderdale, FL). Wounds were then infected with 10 µL of 10⁶ colony-forming units (CFUs) of clinical isolates of MRSA IDRL-6169 and/or *P. aeruginosa* IDRL-11442, an isolate with "difficult-to-treat" resistance, suspended in 0.9% sterile saline. MRSA IDRL-6169 is a methicillin- and mupirocin-resistant isolate from a prosthetic hip. *P. aeruginosa* IDRL-11442 is a wound isolate resistant to piperacillin/tazobactam, cefepime, ceftazidime, meropenem, aztreonam, ciprofloxacin, and levofloxacin (22). Bacterial suspensions were permitted to settle in wound beds for 5 min. Subsequently, the wounds were covered with semi-occlusive transparent Tegaderm (3M, St. Paul, MN) secured using the liquid adhesive Mastisol (Eloquest Health care, Ferndale, MI). Images of the wounds were captured, and wound diameters documented every other day by using a Silhouette wound imaging system (Aranz Medical Ltd, Christchurch, NZ). Purulence was assessed before and after treatment to evaluate immune response to biofilm infection and treatment. The purulence scoring system uses the following scale: 0, no exudate in the wound bed; 1, slight turbid exudate at the wound site; 2, mild amount of white exudate at the wound site; 3, moderate amount of white exudate at the

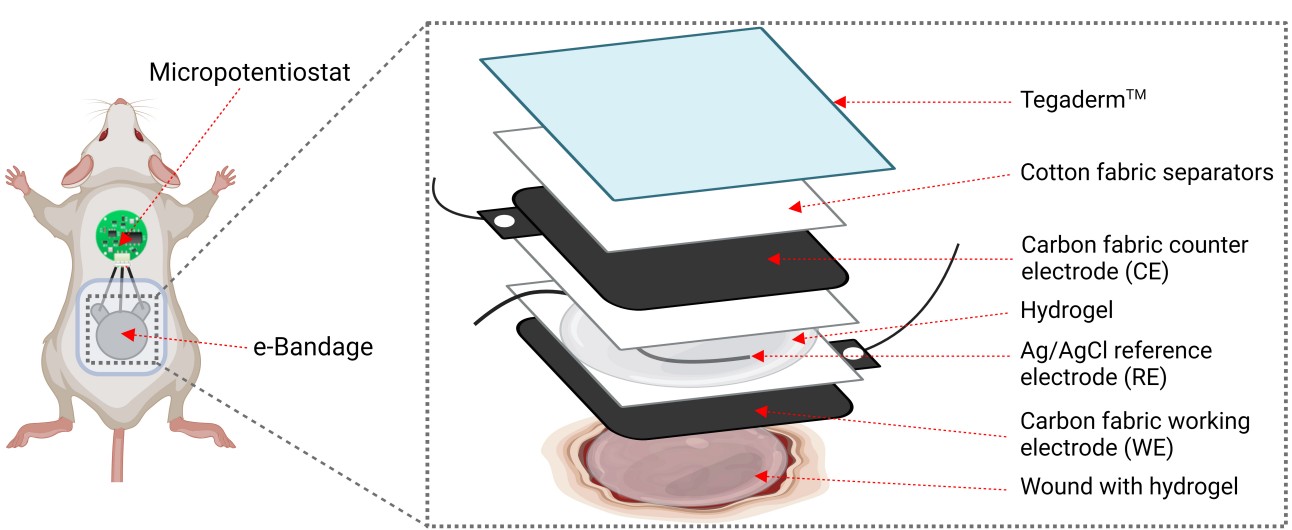

**FIG 1** Schematic of a wounded mouse with e-bandage affixed.

wound site; 4, moderate amount of yellowish exudate at the wound site; 5, large amount of turbid yellow exudate extending beyond the wound bed (20).

## e-Bandage treatment

Following the establishment of 48-h infections in mouse wound beds, mice were anesthetized with isoflurane, Tegaderm was removed, and potentiostats were sutured to the scruff of the neck. Sterile e-bandages were pre-hydrated in sterile 1× phosphate-buffered saline (1× PBS), and 200 µL of sterile hydrogel (1.8% [w/v] xanthan gum in 1× PBS) was injected between the e-bandage layers. An additional 200 µL of hydrogel was applied to the wound beds, and e-bandages were sutured on top to maintain close contact of the entire WE with the dorsal surface during mouse activity. The e-bandages were then connected to the potentiostats, and an additional 200 µL of hydrogel was placed on top, after which the entire e-bandage setups were covered with Tegaderm. Coin cell batteries (3V, Ecr1220 Energizer, St. Louis, MO) were inserted into the mciropotentiostats to initiate e-bandage polarization (HOCl production). Treatment was administered for 48-h with hydrogel refreshment and battery changes every 24-h. Potentials of the WEs relative to the REs were measured following treatment initiation, before and after each battery change, and before euthanasia to continuous operation.

Control groups included wounds administered only hydrogel and Tegaderm, and wounds treated with non-polarized e-bandages (i.e., no potentiostat or HOCl production). Additional animals from the experimental and control groups underwent concurrent antibiotic dosing, with MRSA-infected mice treated with vancomycin and MRSA plus *P. aeruginosa*-infected mice treated with vancomycin and amikacin. Previously, the pharmacokinetic profiles of amikacin and vancomycin were established in Swiss Webster mice to determine a treatment dose of 15 mg/kg subcutaneously administered every 6 h for amikacin and 150 mg/kg intraperitoneally administered every 12 h for vancomycin (21). At least seven mice each were included in the experimental and control groups.

## Wound biofilm quantification

Following treatment, Tegaderm and e-bandages were removed from wound beds, and wound tissue was excised using a 10-mm biopsy punch tool (Acuderm Inc., Fort Lauderdale, FL). The skin tissue was weighed, homogenized (Omni International, Kennesaw, GA) in sterile PBS, vortexed for 20 s , and sonicated for 5 min in a water bath. Subsequently, 100 µL of the resulting homogenate underwent serial dilution (10-fold dilutions) in 0.9% saline, and CFUs were determined by spread-plating 100 µL of each dilution onto tryptic soy agar with 5% sheep blood. Enumeration of dual-species biofilm CFU counts was conducted using eosin methylene blue and colistin nalidixic acid agar plates to select for *P. aeruginosa* and MRSA, respectively. After 24-h of incubation at 37℃, the colonies were counted, and results were reported as $\log_{10}$ CFU/g of tissue.

## Total wound HOCl measurement

Following wound bed excision and homogenization, the remaining portion (900 µL) of the wound homogenate, which was not used for quantifying bacterial load, was utilized to determine the total wound HOCl content by using free-chlorine spectrophotometer test kits (TNT866; Hach Company, Ames, IA), following the manufacturer's instructions. In brief, homogenized wound contents were mixed with 4.1 mL of 1× PBS and centrifuged at 5000 rcf for 15 min. The resulting supernatants were filtered through syringe filters (0.22-µm pore size), and 4 mL of the filtrate was added to free-chlorine test tubes, allowing them to react for 1 min before being measured at 515 nm using a Hach DR 1900 portable spectrophotometer (Hach Company). The free-chlorine content was then converted to HOCl content using a specific equation, considering volume adjustment.

$$\text{Concentration of HOCl } [\text{M}] = \frac{\text{Concentration of free chlorine } \left[\frac{\text{mg}}{\text{L}}\right] \times \text{ conversion factor } \left[\frac{\text{g}}{\text{mg}}\right]}{\text{Molecular weight of chlorine } \left[\frac{\text{g}}{\text{mol}}\right]}$$

The molecular weight of chlorine (70.906 g/mol) and a conversion factor of 0.001 g/mg were used. Complete conversion of HOCl from free chlorine was assumed.

## Histopathology

For the treatment and control groups, a subset of animals ($n$ = 3) was utilized for wound histopathology evaluation. The wounds were excised using a 10-mm biopsy punch and preserved in 10% formalin. After fixation, the specimens were stained with hematoxylin and eosin (H&E). Subsequently, a board-certified clinical pathologist, who was not aware of the sample treatment group, examined the slides. The pathologist assessed the level of inflammation on a scale from 0 (none) to 3 (severe), and checked for the presence of abscesses, ulceration, tissue death, and neutrophil infiltration, marking each as either present or absent.

## Scanning electron microscopy

Following e-bandage treatment, the wound tissues from a subset of three animals from the treatment and control groups were extracted with a 10-mm biopsy punch (Acuderm Inc., Fort Lauderdale, FL) and placed in sterile tubes containing a fixative solution composed of 4% formaldehyde plus 1% glutaraldehyde in phosphate buffer. The samples were then rinsed in PBS and dehydrated through a series of ethanol washes (10%, 30%, 50%, 70%, 90%, 95%, and 100% twice). The dehydrated samples underwent critical point drying in a vacuum sputter coater (Bio-Rad E5100) and were coated with gold/palladium (60%/40%). Finally, samples the were visualized using a Hitachi S4700 cold-field emission scanning electron microscope (Hitachi High Technologies America, Inc., Schaumburg, IL). The samples were assigned non-descriptive numbers upon collection by study staff and randomized by an electron microscopy technologist before imaging. The images were blindly reviewed by eight members of the Mayo Clinic Infectious Diseases Research Laboratory and scored on a scale of 1–3 as follows: 1) *Biofilm matrix integrity/abundance* – EPS was scored on a scale from 1 to 3, with 3, 2, and 1 indicating high, medium, and low abundance, respectively. 2) *Bacterial cell integrity* – microbial cell integrity was assessed by identifying manifestations of cellular distress or damage, including shortening, compacting, dimpling, blistering, and active lysis. Scoring for microbial cell integrity was based on the percentage of cells exhibiting these features: a score of 3 was given when ≥25% of the cells showed signs of damage, a score of 2 when 10%–25% of the cells were affected, and a score of 1 when ≤10% of the cells displayed these features. 3) *Bacterial cell abundance* – Bacterial cell abundance was scored after assessing EPS abundance by counting the number of cells per field of view at 10,000× magnification. A high score of 3 was given when there were ≥75 cells per field of view in low-density EPS or when cells were visibly prevalent throughout high-density EPS. A medium score of 2 was assigned when there were 25–75 cells per field of view in low-density EPS or when cells were prevalently associated with medium-density EPS or sparsely associated with high-density EPS. A low score of 1 was given when there were ≤25 cells per field of view in low-density EPS or when cells were sparsely associated with medium- or high-density EPS. For all parameters, the scoring system utilized objective impressions of the reviewers (after being provided a standardized set of instructions, Supplemental Material 1) from 3 to 4 fields of view per sample (captured from de-identified samples by the microscopist with no knowledge of treatment group or infection type).

## Toxicity screen analysis and inflammatory panel screening

After euthanasia, blood was drawn via cardiac puncture and then centrifuged to separate plasma. The Plasma samples were then examined for various biochemical markers using

a Piccolo Xpress Chemistry Analyzer at the Mayo Clinic Central Clinical Laboratory. This analysis included measuring levels of glucose, amylase, blood urea nitrogen, alkaline phosphatase, alanine aminotransferase, aspartate aminotransferase, gamma glutamyl-transferase, lactate dehydrogenase, C-reactive protein, total bilirubin, creatinine, uric acid, albumin, total protein, calcium, chloride, magnesium, potassium, sodium, and total carbon dioxide. Furthermore, the plasma was analyzed with a MesoScale Discovery SQ 120 to determine the presence of inflammatory biomarkers, including interferon gamma (IFN-γ), interleukin 4 (IL-4), interleukin 5 (IL-5), interleukin 6 (IL-6), tumor necrosis factor alpha (TNF-α), and keratinocyte chemoattractant/human growth-regulated oncogene (KC/GRO).

## Statistical analysis

SEM scores from blind review were compared using ordinary two-way analysis of variance (ANOVA) with Tukey's multiple comparisons test, with a single pooled variance. This allowed for comparison of pooled reviewer scores for all sample types while accounting for reviewer and sample variability within each treatment group. Histopathological profiles were compared with Fisher's exact test. For all other parameters, initial analysis among the experimental groups was performed using the Kruskal–Wallis test. For further detailed comparisons between specific groups, the Wilcoxon rank sum test was applied. The choice of non-parametric tests was driven by the small size of the samples and the lack of evidence supporting the normal distribution of the data. All statistical tests were conducted as two-tailed, considering $P < 0.05$ as statistically significant. When dealing with comparisons involving more than three groups, adjustments were made to account for the false discovery rate. The data analysis was conducted using SAS software (version 9.4, SAS Institute), whereas GraphPad Prism (version 10.1, GraphPad Software) was used for creating graphs.

## RESULTS

### HOCl was produced by polarized e-bandages *in situ*

In previous studies, microelectrodes were used to demonstrate that e-bandages generate HOCl at the working electrode. HOCl was shown to penetrate biofilms, explant tissue, and wound beds in live mice (14, 15, 23). In this study, free-chlorine spectropho-tometer test kits were used to quantify concentrations of HOCl in wounds infected with MRSA alone, and combined with *P. aeruginosa*. In both infection scenarios, wounds from mice treated with polarized e-bandages exhibited elevated levels of HOCl compared with those treated with non-polarized e-bandages or Tegaderm alone (Fig. 2).

### Wound bacterial loads were reduced by polarized e-bandage treatment

To assess the efficacy of HOCl-generating e-bandage therapy in reducing bacterial biofilm burden *in vivo*, endpoint wound CFUs were quantified after 48-h of treatment. Treatment of MRSA wound biofilms with polarized e-bandages reduced bacterial loads compared with non-polarized e-bandages ($P = 0.0048$) or Tegaderm alone ($P = 0.0048$, Fig. 3A). Treatment of wound biofilms infected with both MRSA and *P. aeruginosa* by polarized e-bandages reduced bacterial loads of both species compared with non-polarized e-bandages (MRSA: $P = 0.026$; *P. aeruginosa*: $P = 0.0032$) or Tegaderm alone (MRSA: $P = 0.003$; *P. aeruginosa*: $P = 0.0015$; Fig. 3B).

SEM images of three wounds from each treatment group for both MRSA alone and MRSA plus *P. aeruginosa* infected mice (Fig. 4B) were blindly scored for biofilm matrix integrity, bacterial cell integrity, and bacterial cell abundance. Bacterial cell abundance was significantly lower after polarized e-bandage treatment for both MRSA and MRSA plus *P. aeruginosa*-infected wounds compared with Tegaderm only and non-polarized control groups (Fig. 4A), in agreement with reduced bacterial loads.

To test if e-bandage treatment of established wound biofilms exhibited synergy with antibiotics against established wound biofilms, additional mice from all groups for both

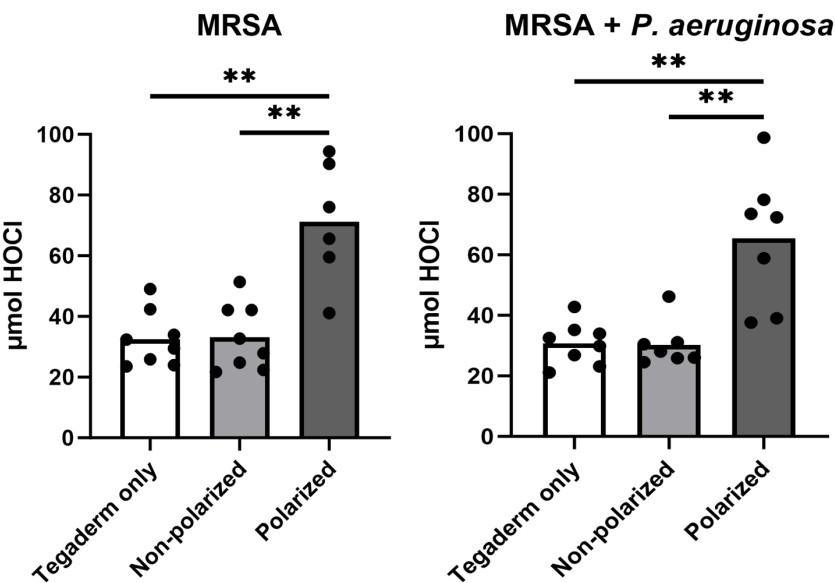

**FIG 2** Polarized e-bandage treatment resulted in increased total wound HOCl content. The 48-h wound bed biofilms containing MRSA or MRSA plus *P. aeruginosa* were treated for 48-h with either polarized (HOCl-producing) or non-polarized e-bandages and compared with Tegaderm only controls. Statistical analysis was performed using the Wilcoxon rank sum test with correction for false discovery rate. Individual data points with the means (bars) are shown. $N \geq 7$. **$P \leq 0.01$.

MRSA and MRSA plus *P. aeruginosa* infections were administered concurrent systemic vancomycin (for MRSA alone), or vancomycin and amikacin (for MRSA plus *P. aeruginosa*) for the duration of the e-bandage treatment. Antibiotic treatment did not result in lower end-point bacterial loads for wounds treated with polarized e-bandages for either the single or dual-species infection groups. For MRSA alone, vancomycin reduced bacterial loads in only the non-polarized group (Fig. 3A). For MRSA plus *P. aeruginosa*, MRSA load was not reduced by the addition of antibiotics with any treatment type; however, *P. aeruginosa* was reduced in the antibiotic-containing groups in mice receiving both Tegaderm only and non-polarized e-bandages (Fig. 3B).

### Wound healing was not hampered by polarized e-bandage treatment

To determine if the HOCl-producing e-bandage treatment, with and without concurrent systemic antibiotics, affected wound closure over 48-h of treatment, the total wound area was measured before and after application. No significant differences in overall wound closure percentage were observed between any non-antibiotic-treated groups or between any antibiotic-treated groups for either MRSA alone or MRSA plus *P. aeruginosa* infections (Fig. 5). Interestingly, wound closure was greater in the non-polarized group for MRSA plus *P*. aeruginosa-infected wounds when antibiotics were used but not in the Tegaderm only group or in wounds infected with MRSA alone.

### Treatment of infected wounds with polarized e-bandages resulted in reduced purulence

The impact of e-bandage and/or antibiotic therapy on wound bed purulence was evaluated by scoring purulence before and after treatment (Fig. 6). Polarized e-bandages resulted in a marked reduction in purulence compared with the Tegaderm only control group in wounds infected with both MRSA and MRSA combined with *P. aeruginosa*. There was no significant improvement in purulence reduction between polarized and non-polarized e-bandage groups for either the mono- or dual-species infection, although the non-polarized group exhibited significantly less purulence than the Tegaderm only group in dual-species infections (and to a lesser, insignificant amount in the MRSA only

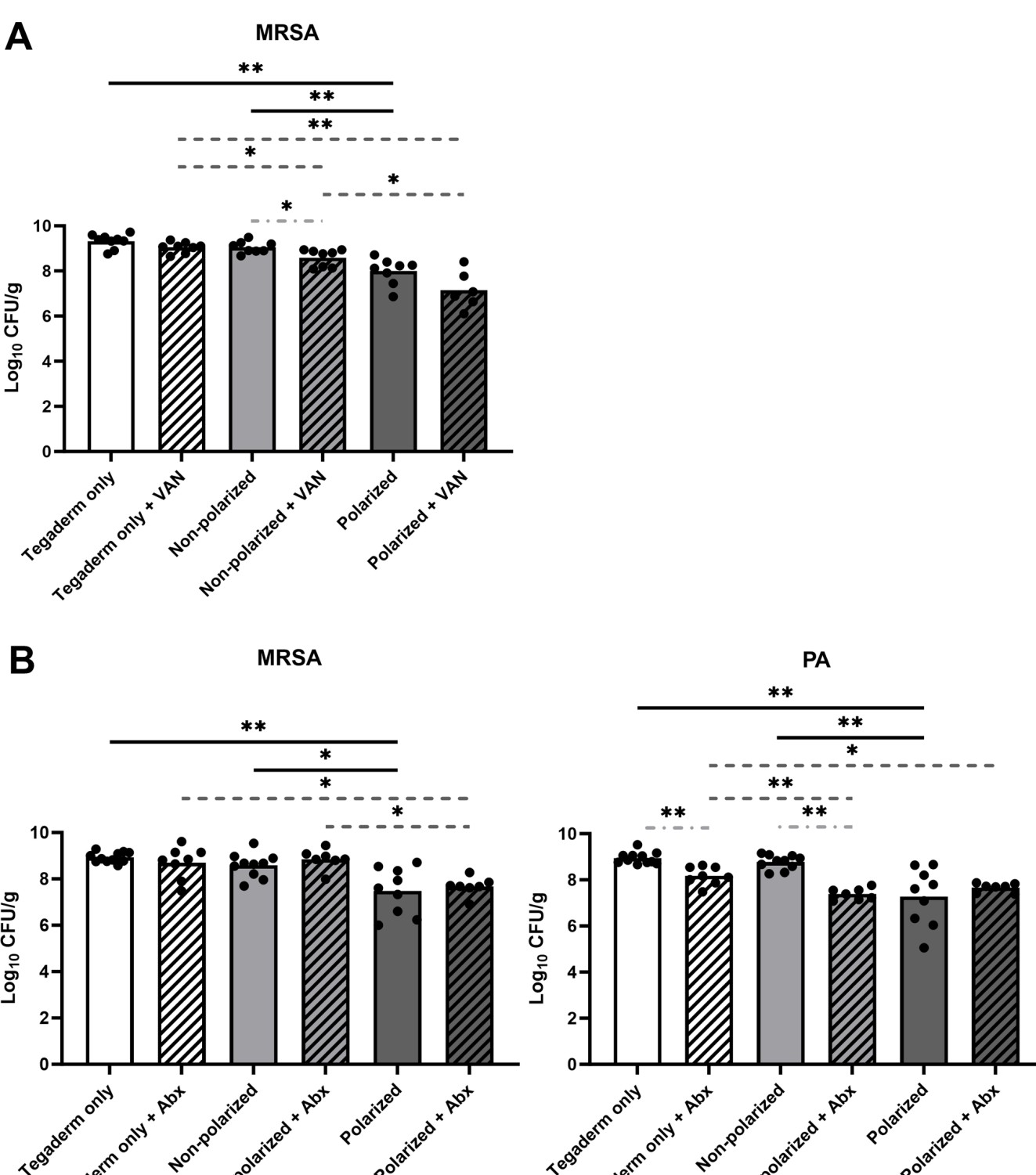

**FIG 3** Polarized e-bandage treatment reduces endpoint bacterial loads. The 48-h wound bed biofilms containing MRSA (A) or MRSA plus *P. aeruginosa* (PA; B) were treated for 48-h with either polarized (HOCl-producing) or non-polarized e-bandages, with or without systemic antibiotics (MRSA alone, vancomycin – VAN; MRSA plus *P. aeruginosa* – vancomycin plus amikacin - Abx) and compared with Tegaderm only controls, with and without antibiotics. Statistical analysis was performed using the Wilcoxon rank sum test with correction for false discovery rate. Individual data points with the means (bars) are shown. Solid black significance bars show differences between non-antibiotic-treated groups; dashed dark grey significance bars show differences between antibiotic-treated groups; light gray dashed and dotted significance bars show differences between antibiotic and non-antibiotic-treated groups with the same e-bandage treatment type. $N \geq 7$. *$P \leq 0.05$, **$P \leq 0.01$.

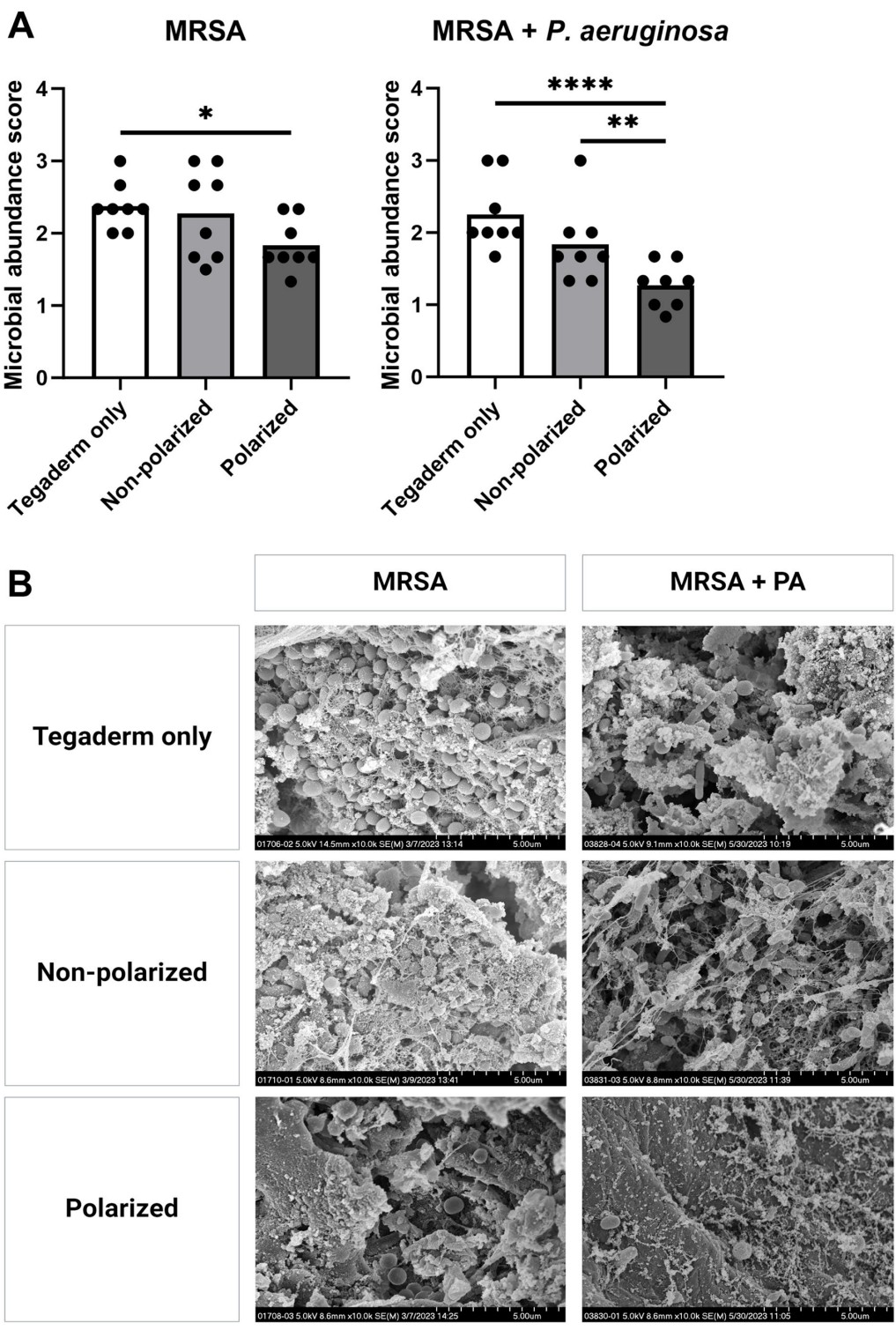

**FIG 4** Polarized e-bandage treatment reduces bacterial abundance observed in scanning electron microscopy (SEM) images. SEM images of 48-h wound biofilms containing MRSA or MRSA plus *P. aeruginosa* (PA) treated for 48-h with polarized or non-polarized e-bandages, or Tegaderm only, were blindly reviewed and scored for bacterial abundance, with the scores plotted in (A). Individual data points with the means (bars) are shown. Representative images are shown in (B) at 10,000× magnification. Statistical significance was determined via two-way ANOVA with Tukey's multiple comparisons test, with a single pooled variance. *N* = 3 samples per treatment type; 3–4 images per sample informed scoring. *$P \leq 0.05$, **$P \leq 0.01$. **$P \leq 0.0001$.

infections). Concurrent antibiotics did not improve purulence reduction in any treatment group in either mono- or dual-species-infected wounds.

## Polarized e-bandage treatment produced no observable tissue toxicity

To ascertain whether e-bandage therapy led to increased tissue toxicity compared with infection alone, the samples were evaluated by a clinical pathologist blinded to the treatment. No notable variances were observed in overall inflammation, necrosis levels, abscess formation, ulceration, or neutrophilic inflammation across all treatment groups for wounds infected with MRSA or MRSA plus *P. aeruginosa* (Table 1).

## Assessment of inflammation and blood biomarkers for indication of animal health

Blood biochemical biomarker assessment and measurement of inflammatory cytokines was performed on a subset ($n = 3$) of animals from each group to examine the immune response and general health of infected animals compared with uninfected control animals at the time of euthanasia. As expected, all infected animals exhibited an elevated proinflammatory response compared with uninfected controls for both infection types (Fig. 7). In particular, the proinflammatory cytokines INF-γ and IL-6 were elevated approximately four- to 16-fold and two- to ninefold, respectively, indicating a strong, macrophage-driven immune response in all infected groups. Between infected groups, only KC/GRO showed significant elevation in animals treated with polarized vs non-polarized e-bandages for MRSA infection alone. Notably, IL-6 was also the most elevated in the polarized group for both infection types, albeit not to the level of statistical significance. For blood biochemical analysis, the mean analyte levels were within normal healthy range for all groups, with no significant difference between animals treated with polarized or non-polarized e-bandages for both infection types (data not shown).

## DISCUSSION

Development of new antimicrobial strategies is imperative in the face of rising antibiotic-resistant bacterial pathogens and is particularly relevant to polymicrobial wound

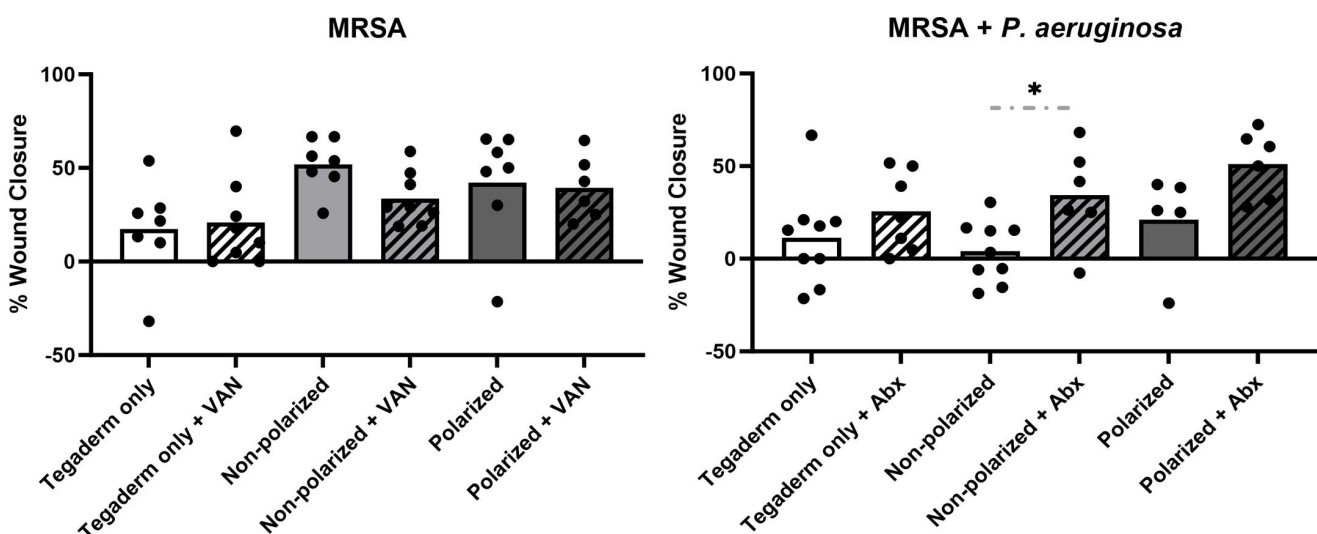

**FIG 5** Polarized e-bandage treatment did not hinder wound closure. The 48-h wound bed biofilms containing MRSA or MRSA plus *P. aeruginosa* were treated for 48-h with either polarized (HOCl-producing) or non-polarized e-bandages, with or without systemic antibiotics (MRSA alone, vancomycin – VAN; MRSA plus *P. aeruginosa* – vancomycin plus amikacin - Abx) and compared with Tegaderm only controls, with and without antibiotics. The wound area was measured before and after treatment. Individual data points with means (bars) are shown. Statistical analysis was performed using the Wilcoxon rank sum test with correction for false discovery rate. $N \geq 7$. *$P \leq 0.05$.

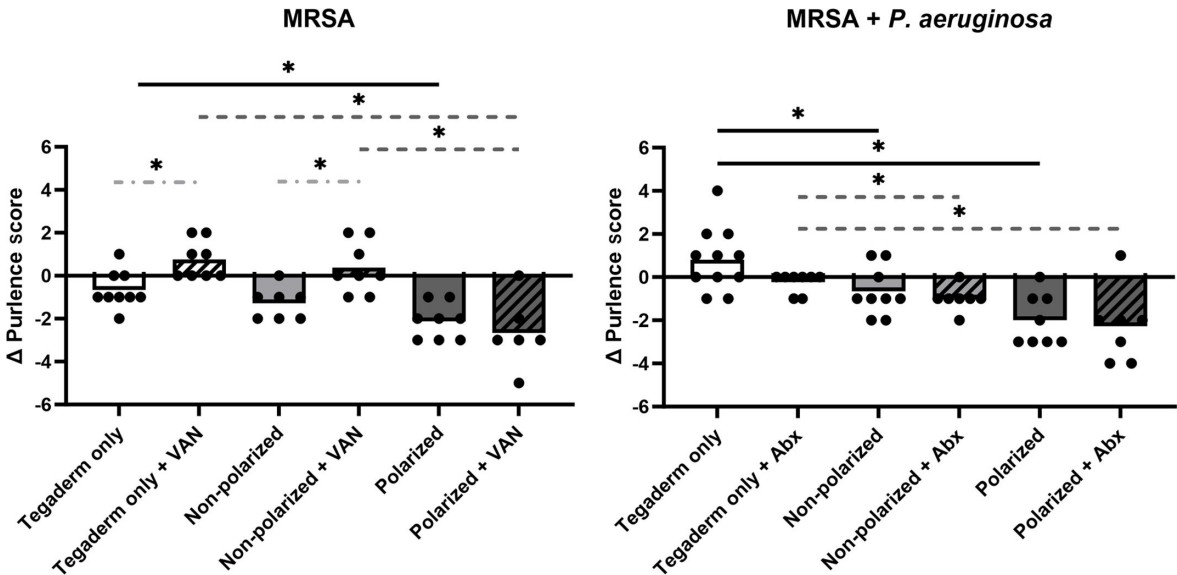

**FIG 6** The e-bandage treatment resulted in reduced wound purulence. The 48-h wound bed biofilms containing MRSA or MRSA plus *P. aeruginosa* were treated for 48-h with either polarized (HOCl-producing) or non-polarized e-bandages, with or without systemic antibiotics (MRSA alone, vancomycin – VAN; MRSA plus *P. aeruginosa* – vancomycin plus amikacin - Abx) and compared with Tegaderm only controls, with and without antibiotics. Wound purulence was scored before and after treatment. Statistical analysis was performed using the Wilcoxon rank sum test with correction for false discovery rate. Individual data points with means (bars) are shown. Solid black significance bars show differences between non-antibiotic-treated groups; dashed dark gray significance bars show differences between antibiotic-treated groups; light gray dashed and dotted significance bars show differences between antibiotic and non-antibiotic-treated groups with the same e-bandage treatment type. $N \geq 7$. *$P \leq 0.05$.

infections. In this study, the efficacy of a previously developed HOCl-producing e-bandage for treatment of wound biofilm infections with antibiotic-resistant clinical isolates of *S. aureus* and *P. aeruginosa* was investigated. In a previous study, the efficacy of HOCl-producing e-bandages against wounds infected with *P. aeruginosa* alone was demonstrated (14). Polymicrobial infections, particularly involving antibiotic resistant strains, pose additional challenges to wound infection healing. MRSA and *P. aeruginosa* are two of the most commonly isolated wound pathogens and are often found together (3, 4), with worse outcomes compared with mono-species infections (24–26).

Results confirm the ability of polarized e-bandages to produce HOCl *in situ*, leading to elevated levels of HOCl in wound beds compared with non-polarized e-bandages or Tegaderm alone. Production of HOCl was associated with a significant reduction in bacterial biofilm burden *in vivo*, as demonstrated by lower bacterial loads in wounds infected with MRSA alone or co-infected with MRSA and *P. aeruginosa* following treatment with polarized e-bandages. Further, blind review of SEM images of the wound beds from all groups revealed lower bacterial abundance in animals treated with polarized e-bandages. Reductions of 1–2 $\log_{10}$ CFU/g, and not more, may be because wounds in this model were mature, with high bacterial bioburden (~$10^9$ $\log_{10}$ CFU/g), and HOCl was delivered at low, continuous concentrations for only 48-h. Previous studies involving

**TABLE 1** Histopathological profiles[a]

| Group | Inflammation (none-0; mild-1; moderate-2; severe-3) | Percent necrosis noted | Percent abscess noted | Percent ulceration noted | Percent neutrophilic inflammation |
|---|---|---|---|---|---|
| Polarized | 3 | 100% | 100% | 100% | 100% |
| Non-polarized | 2.83 | 100% | 83% | 100% | 83% |
| Tegaderm only | 3 | 100% | 100% | 100% | 100% |

[a]Sections of 48-h wound infections treated with either polarized e-bandage, non-polarized e-bandage, or Tegaderm only for 48-h were blindly scored by a clinical pathologist. $N = 6$ per group (MRSA and MRSA + *P. aeruginosa* combined, $N = 3$ each per group). No significant differences were noted between any group (Fisher's exact test).

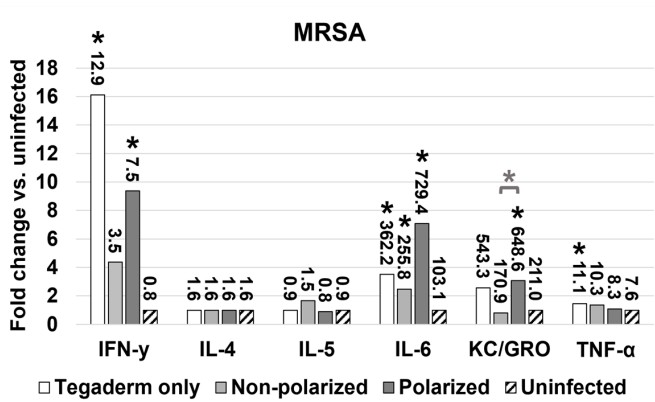
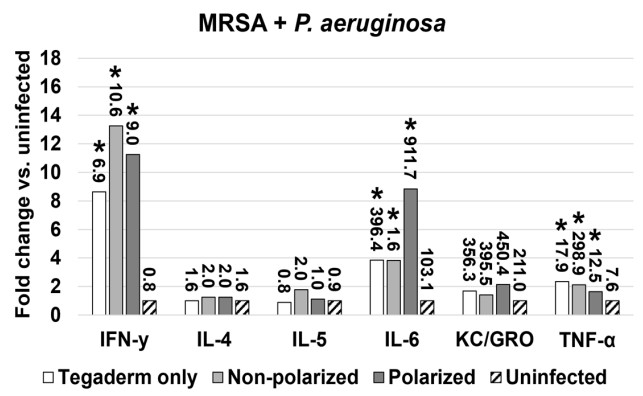

**FIG 7** Inflammatory response in infected groups compared with uninfected controls. The 48-h wound bed biofilms containing MRSA or MRSA plus *P. aeruginosa* were treated for 48-h with either polarized (HOCl-producing) or non-polarized e-bandages and compared with Tegaderm only controls. Following treatment, plasma was collected and analyzed for the levels of IFN-γ, IL-4, IL-5, IL-6, TNF-α, and KC/GRO. Fold change in comparison to uninfected controls is shown, with analyte levels (in pg/mL) displayed as data labels. Statistical analysis was performed using the Wilcoxon rank sum test with correction for false discovery rate. Asterisks without bars represent significance compared with uninfected controls. Asterisks with bars represent significance between groups for the same analyte. $N = 3$. *$P \leq 0.05$.

HOCl typically involved less mature infections, with lower bioburden, and/or were treated for longer durations. For example, Kuwabara et al. showed that mouse wounds infected with ~$10^6$ $\log_{10}$ CFU/g of *P. aeruginosa* were reduced by roughly $10^3$ to $10^4$ $\log_{10}$ CFU/g when treated with HOCl combined with chitin-nanofiber sheets or immobilized silver nanoparticles for 12 days (27). Robson et al. found that stabilized HOCl treatment of murine wounds infected with biofilms containing ~$10^7$ to ~$10^9$ CFU/g *P. aeruginosa* resulted in reductions of between 3 and 6 $\log_{10}$ CFU/g after 20 days of treatment, depending on the regimen (28).

No significant effect on biofilm matrix was observed, indicating that the treatment is likely directly biocidal to biofilm-dwelling pathogens, as opposed to acting as an anti-EPS or pro-dispersal agent. HOCl is a strong oxidizing agent that damages microbial cells by interacting with membrane lipids, nucleic acids, proteins, sulfur-containing amino acids, and more (29, 30); description of effects against biofilm EPS, especially *in vivo*, is lacking.

Although polarized e-bandage treatment alone effectively reduced bacterial loads, addition of systemic antibiotics did not result in additional microbicidal activity for either the MRSA-infected or MRSA plus *P. aeruginosa*-infected wounds, indicating that the antibacterial efficacy of e-bandages is independent of systemic antibiotic administration. This highlights the potential of e-bandages as a standalone antimicrobial strategy for wound infections. Notably, for both MRSA and *P. aeruginosa*, polarized e-bandage treatment combined with antibiotics was more effective than antibiotics alone, indicating a lack of antagonistic effects between the generated HOCl and the antibiotics. Additionally, for wounds infected with MRSA alone, vancomycin treatment resulted in significantly more reduction to bacterial load in the non-polarized e-bandage control group but not in the Tegaderm only control group. Although the reason for this is unclear, one possible explanation is that the presence of the e-bandage could be physically disturbing the biofilm matrix, leading to increased antibiotic penetrance.

All infected groups showed an elevated, macrophage-driven immune response compared with uninfected controls. Between the infected groups, compared with non-polarized e-bandage controls, the animals treated with polarized e-bandages showed significantly elevated levels of KC/GRO when infected with MRSA alone, and potentially elevated levels of IL-6 (though statistically insignificant) when infected with both MRSA alone and in combination with *P. aeruginosa*. KC/GRO acts as a chemo-attractant for neutrophils and other phagocytes, playing a role in the acute inflammatory response (31), whereas IL-6 is a cytokine produced by various immune cells in response to infection and tissue injury, promoting inflammation and stimulating

immune responses (32). Elevation of both suggests that inflammation in the polarized group may be more pronounced, possibly due to the treatment itself (albeit unlikely, as HOCl is produced below toxic levels) (10) or increased recognition of pathogens (or pathogen debris) by the immune system.

No adverse effects on wound healing or tissue toxicity associated with polarized e-bandage treatment was observed. Assessment of wound closure, purulence, histopathology, and blood biomarkers revealed no differences between non-polarized and polarized groups, indicating the safety and biocompatibility of e-bandage in this context. Notably, purulence was higher in the antibiotic-treated groups for MRSA-infected wounds treated with both Tegaderm only and non-polarized e-bandages. Although the reason for this is unclear, one possibility is that the presence of vancomycin-lysed bacteria stimulates the innate immune response. Additionally, although wound closure was not improved by polarized e-bandage treatment, it is possible that a treatment duration longer than 48-h could yield a difference as there was a trend towards improved closure, especially in the dual-species infection groups.

Previous results with e-bandages that produce an alternative reactive oxygen species, hydrogen peroxide ($H_2O_2$), found that wound healing was augmented (21); however, the antimicrobial efficacy of electrochemically generated $H_2O_2$ was less than HOCl against a broad spectrum of microorganisms (12, 16, 23, 33–35). Therefore, a programmable e-bandage that can produce both HOCl and $H_2O_2$ for optimal biocide and wound-healing augmentation, respectively, should be explored.

In conclusion, these findings support the promising efficacy of polarized HOCl-producing e-bandages in treating wound biofilm infections containing MRSA and *P. aeruginosa*. The ability of e-bandages to locally generate HOCl offers a novel antimicrobial strategy that addresses challenges associated with antibiotic resistance in wound management, particularly in the context of polymicrobial infections. Further studies are warranted to validate these findings and assess clinical application of e-bandage therapy for the treatment of wound infections.

## ACKNOWLEDGMENTS

Research reported in this publication was supported by the National Institute of Allergy and Infectious Diseases of the National Institutes of Health under award number R01AI091594. The content is solely the responsibility of the authors and does not necessarily represent the official views of the National Institutes of Health.

## AUTHOR AFFILIATIONS

[1]Division of Clinical Microbiology, Mayo Clinic, Rochester, Minnesota, USA
[2]The Gene and Linda Voiland School of Chemical Engineering and Bioengineering, Washington State University, Pullman, Washington, USA
[3]Department of Quantitative Health Sciences, Mayo Clinic, Rochester, Minnesota, USA
[4]Division of Public Health, Infectious Diseases, and Occupational Medicine, Mayo Clinic, Rochester, Minnesota, USA

## AUTHOR ORCIDs

Derek Fleming  http://orcid.org/0000-0002-0054-904X
Md Monzurul Islam Anoy  http://orcid.org/0000-0001-6876-2632
Audrey N. Schuetz  http://orcid.org/0000-0002-5837-270X
Haluk Beyenal  http://orcid.org/0000-0003-3931-0244
Robin Patel  http://orcid.org/0000-0001-6344-4141

## FUNDING

| Funder | Grant(s) | Author(s) |
|---|---|---|
| HHS | NIH | National Institute of Allergy and Infectious Diseases (NIAID) | R01AI091594 | Derek Fleming |
| | | Melissa J. Karau |
| | | Kerryl E. Greenwood-Quaintance |
| | | Haluk Beyenal |
| | | Robin Patel |

## AUTHOR CONTRIBUTIONS

Derek Fleming, Conceptualization, Data curation, Formal analysis, Investigation, Methodology, Writing – original draft, Writing – review and editing | Ibrahim Bozyel, Conceptualization, Investigation, Methodology, Writing – review and editing | Christina A. Koscianski, Investigation, Methodology, Writing – review and editing | Dilara Ozdemir, Investigation, Methodology, Writing – review and editing | Melissa J. Karau, Investigation, Methodology, Writing – review and editing | Luz Cuello, Conceptualization, Investigation, Methodology, Writing – review and editing | Md Monzurul Islam Anoy, Conceptualization, Investigation, Methodology, Writing – review and editing | Audrey N. Schuetz, Conceptualization, Formal analysis, Investigation, Methodology, Supervision, Writing – review and editing | Kerryl E. Greenwood-Quaintance, Conceptualization, Formal analysis, Methodology, Supervision, Writing – review and editing | Jayawant N. Mandrekar, Conceptualization, Formal analysis, Funding acquisition, Methodology, Writing – review and editing | Haluk Beyenal, Conceptualization, Formal analysis, Funding acquisition, Methodology, Supervision, Writing – review and editing | Robin Patel, Conceptualization, Funding acquisition, Investigation, Methodology, Supervision, Writing – review and editing.

## ADDITIONAL FILES

The following material is available online.

### Supplemental Material

**Supplemental material (Spectrum00626-24-s0001.pdf).** Guide to scoring of *in vivo* biofilms from SEM images.

### Open Peer Review

**PEER REVIEW HISTORY (review-history.pdf).** An accounting of the reviewer comments and feedback.

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
