## [Reviewer comments · Microbiology Spectrum]

Microbiology Spectrum

HOCl-producing Electrochemical Bandage is Active in Murine Polymicrobial Wound Infection

Derek Fleming, Ibrahim Bozyel, Christina Koscianski, Dilara Ozdemir, Melissa Karau, Luz Cuello, Md Anoy, Suzanne Gelston, Audrey Schuetz, Kerryl Greenwood-Quaintance, Jayawant Mandrekar, Haluk Beyenal, and Robin Patel

Corresponding Author(s): Robin Patel, Mayo Clinic Minnesota

Review Timeline:

Submission Date:	March 8, 2024
Editorial Decision:	May 23, 2024
Revision Received:	June 24, 2024
Accepted:	July 1, 2024

Editor: Cristina Solano

Reviewer(s): Disclosure of reviewer identity is with reference to reviewer comments included in decision letter(s). The following individuals involved in review of your submission have agreed to reveal their identity: Sarah E. Maddocks (Reviewer #1)

Transaction Report:

DOI: <https://doi.org/10.1128/spectrum.00626-24>

Re: Spectrum00626-24 (HOCl-producing Electrochemical Bandage is Active in Murine Polymicrobial Wound Infection)

Dear Dr. Robin Patel:

Thank you for the privilege of reviewing your work. Below you will find my comments, instructions from the Spectrum editorial office, and the reviewer comments.

You will see that the reviewers, although they found merit in your study, have raised a number of concerns that preclude acceptance in its present form. If you feel able to address these comments and the points mentioned below and in the attached document, then I will be happy to consider a revised manuscript.

Revision Guidelines

Sincerely,
Cristina Solano
Editor
Microbiology Spectrum

Reviewer #1 (Comments for the Author):

This is an interesting and well written manuscript. The approach uses robust methods, but am not sure the statistical analyses is correct. For some data there seem to be clear differences but no apparent significance.

Other comments below:

The abstract should be written more concisely and give an clear summary of the study, at present it gives no indication of the methodological approach, and the findings are not present in a clear manner.

In the introduction (lines 64-76) some of this methodology would be better in the abstract.

The methods describe the use of EMB and naladixic acid plats to recover bacteria. Which plates are selective for which organism? This is especially important to note for the mixed-biofilms.

Single species MRSA biofilms were cultured as mixed *S. aureus*/*P. aeruginosa*. Why did the authors not include single species *P. aeruginosa* as a comparison also?

Lines 241-245 please explain why it is likely that the reductions in CFU were observed for the Tegaderm and non-polarised dressings.

Lines 303-304 - how do the authors propose HOCl is directly biocidal to bacteria embedded in a biofilm of it does not have EPS disruptive activity?

306 - the combination of vancomycin with the polarised dressing seemed to decrease the activity of the vancomycin - could the authors indicate why this might be? Have the authors undertaken any in vitro synergy/antagonism assays? As presented it looks like HOCl might impair the antibiotics.

In lines 240-245 - it needs to be clear what "reduced" is referring to. At present it reads as though the tegaderm and non-polarised dressings work well.

I am curious about he results. The log reductions presented are between 1-2. There have been a number of other studies with HOCl that demonstrate much larger log reductions - it would be good for the authors to discuss this and why their data might be different.

On several of the figures, in some cases the differences highlighted appear negligible, but stated as significant, whereas larger differences do not appear to be significant (e.g. fig 4). Have the authors used the correct analysis? Could the calculated SEM be included in the results?

Fig 6 is presented as fold changes - more typically these data are presented as absolute quantities, and I think the finding would be stronger if presented this way.

Comments to the authors:

Fleming et al. use an electrochemical bandage producing HOCl to treat murine wound healing infections containing *P. aeruginosa* and *S. aureus* MRSA. The authors compared the polarized e-bandage producing HOCl treatment to a non-polarized e-bandage and to a transparent film dressing only (Tegaderm) with and without antibiotics. They analyzed HOCl production, CFUs reduction, wound closure, reduction of purulence, tissue toxicity and inflammation, reporting the efficacy of polarized e-bandages in treating wound biofilm infections. The overall work and results are relevant; however, I would like to express some concerns about the interpretation and discussion of the results presented.

Major concerns about results interpretation:

1. Figure 3B is not commented in the results section. Could the authors please address this?
2. In the results section, the authors state that “No significant differences in overall wound closure percentage were observed” (line 249). However, in Figure 4, there is a significant increase in wound closure percentage in the Non-polarized + Abx group compared to Non-polarized in the co-infection. Please clarify or correct if there is an error.
3. The authors state that “wound closure was less complete in the non-polarized group for MRSA plus *P. aeruginosa*-infected wounds when antibiotics were used” (line 251). However, Figure 4 shows an increase in wound closure when antibiotics are used in the co-infected group. Please explain or amend this statement if incorrect.
4. In lines 253-254 it is stated “This effect (though not significant) was also observed with the Tegaderm only and polarized groups for the dual-infection wounds, but with MRSA alone”. Since the lines above were not clear, this statement is also not clear. Please provide a clearer explanation.
5. In the results section, the authors state that “the non-polarized group exhibited significantly less purulence than the polarized group in the dual-species infections” (line 262). However, in Figure 5 shows a more negative purulence score change in the polarized group. Is there some data missing? Please clarify
6. The results mention “and to a lesser, insignificant amount in the MRSA only infections” (line 263). Where is this shown? Could the authors please explain it better?
7. For the statement “Polarized e-bandage treatment produced no observable tissue toxicity” (lines 266-271), is there supporting data? Please provide it.
8. The authors state “Between infected groups, only KC/GRO showed significant elevation in animals treated with polarized vs non-polarized e-bandages” (lines 280-281). However, Figure 6 shows this is only true for the MRSA alone infection. Please clarify.
9. In Figure 6 legend it mentions plasma collected from animals treated with and without antibiotics. Where can this difference in treatment be seen in the graph? Please clarify.
10. In all the figures, could please the authors include error bars?

In general, there are some results seen in the graphs that are missing discussion. Concerns about discussion section:

11. In the discussion, lines 313-316 it states, "Between infected groups, animals treated with polarized e-bandages showed significantly elevated levels of KC/GRO when infected with MRSA alone, and insignificantly elevated levels of IL-6 when infected with both MRSA alone and in combination with *P. aeruginosa*." Is this "significantly" or "insignificantly" compared to non-polarized? Please specify.
12. Is there previous data on inflammation supporting the findings on the inflammatory response? Could the authors please discuss it?
13. Although non-polarized bandages do not show significantly improvement compared to Tegaderm only, significant improvement is seen when antibiotics are used (decreased CFUs, increased wound closure, reduced purulence). Could the authors please discuss why non-polarized bandages show some improvement in wound healing when antibiotics are used compared to Tegaderm with antibiotics?
14. Could the authors please discuss why there is an increase in purulence when antibiotics are added?
15. Could the authors comment on the fact that if there is no difference in wound closure or purulence between non-polarized and polarized e-bandages, how do the latter help in wound healing, apart from the reduced bacterial load? (lines 319-321)

Minor concerns:

16. In the Methods and Materials section, it is difficult to visualize the electrochemical bandage and the treatment. Could the authors include an image or scheme to better understand them?
17. In the Methods and Materials section, consider moving "Wound biofilm quantification" before "total wound HOCl measurement", to better follow the procedure.
18. In line 163, could the authors specify which bacteria is each selective plate for?
19. In line 186, it is unclear how were SEM images scored on bacterial abundance, cell and matrix integrity. Could the authors explain which parameters were used?
20. Lines 32, 33 and 34: "vs." is missing a point.
21. Lines 46-48: This sentence is difficult to understand, please rephrase.
22. Line 50: DNA in the biofilm extracellular matrix is normally referred to as eDNA.
23. Line 103: Please be consistent with nomenclature (μL instead of μl)
24. Line 104: "difficult to treat" refers to both PA and MRSA? Then maybe it should say have instead of has.
25. Line 140: please state what "IP" refers to.

We thank the reviewers for their critiques of our manuscript. Changes have been made in response to the reviews, as detailed below and in the marked-up version of the document.

Reviewer #1 (Comments for the Author):

This is an interesting and well written manuscript. The approach uses robust methods, but am not sure the statistical analyses is correct. For some data there seem to be clear differences but no apparent significance.

Thank you for your interest and effort. Concerns about statistics have been addressed in response to your last comment.

Other comments below:

The abstract should be written more concisely and give an clear summary of the study, at present it gives no indication of the methodological approach, and the findings are not present in a clear manner.

The abstract has been rewritten for clarity.

In the introduction (lines 64-76) some of this methodology would be better in the abstract.

Methodology has been added to the abstract.

The methods describe the use of EMB and naladixic acid plats to recover bacteria. Which plates are selective for which organism? This is especially important to note for the mixed-biofilms.

This information has been added to the Methods section (Lines 168-169).

Single species MRSA biofilms were cultured as mixed *S. aureus*/*P. aeruginosa*. Why did the authors not include single species *P. aeruginosa* as a comparison also?

Wounds infected with *P. aeruginosa* alone were studied in a prior study. Additional text has been added to the abstract and introduction (Lines 78-79) to make this point clear.

Lines 241-245 please explain why it is likely that the reductions in CFU were observed for the Tegaderm and non-polarised dressings.

This has been addressed in the revised manuscript (Lines 366-370).

Lines 303-304 - how do the authors propose HOCl is directly biocidal to bacteria embedded in a biofilm if it does not have EPS disruptive activity?

This has been addressed in the revised manuscript (Lines 356-358).

306 - the combination of vancomycin with the polarised dressing seemed to decrease the activity of the vancomycin - could the authors indicate why this might be? Have the authors undertaken any in vitro synergy/antagonism assays? As presented it looks like HOCl might impair the antibiotics.

We respectfully disagree with this observation. Polarized + vancomycin (and amikacin) counts were lower than vancomycin alone for both MRSA alone and MRSA + *P. aeruginosa* (dashed significance bars in figure 2). We have added additional text to the Discussion to clarify this point (Lines 363-366).

In lines 240-245 - it needs to be clear what "reduced" is referring to. At present it reads as though the Tegaderm and non-polarised dressings work well.

Clarification has been added (Lines 283-286).

I am curious about the results. The log reductions presented are between 1-2. There have been a number of other studies with HOCl that demonstrate much larger log reductions - it would be good for the authors to discuss this and why their data might be different.

Text has been added to the discussion to address this (Lines 343-353).

On several of the figures, in some cases the differences highlighted appear negligible, but stated as significant, whereas larger differences do not appear to be significant (e.g. fig 4). Have the authors used the correct analysis? Could the calculated SEM be included in the results?

As the reviewer points out, this has to do with variable SEM between the groups, and with corrections for false discovery. That is, for all analyses except electron microscopy scores (for which two-way Anova was used to account for variability between the scorers), the Wilcoxon rank sum test was used in combination with the Benjamini-Hochberg, also known as False Discovery Rate (FDR) procedure. This was done to account for multiple comparisons to control the expected proportion of false discoveries. Traditional approaches for controlling error rates in the presence of multiple comparisons include strong and weak control of familywise error rates, using techniques such as the Bonferroni correction. The false discovery rate approach is more powerful than methods like the Bonferroni correction that control false positive rates.

Fig 6 is presented as fold changes - more typically these data are presented as absolute quantities, and I think the finding would be stronger if presented this way.

Thank you for this suggestion, the data was presented as fold change in order to capture all parameters on the same graph (same y-axis range). However, in consideration of this, the absolute quantities (in mean analyte levels) are included as data labels above each bar.

Reviewer 2

1. Figure 3B is not commented in the results section. Could the authors please address this?

Figure 3b (now 4b) has now been referenced, and clarifying text added (Line 270).

2. In the results section, the authors state that "No significant differences in overall wound closure percentage were observed" (line 249). However, in Figure 4, there is a significant increase in wound closure percentage in the Non-polarized + Abx group compared to Non-polarized in the co-infection. Please clarify or correct if there is an error.

Thank you for this observation; this was our error and has been corrected (Lines 291-292).

3. The authors state that "wound closure was less complete in the non-polarized group for MRSA plus *P. aeruginosa*-infected wounds when antibiotics were used" (line 251). However, Figure 4 shows an increase in wound closure when antibiotics are used in the co-infected group. Please explain or amend this statement if incorrect.

Thank you for this observation; this was our error and has been corrected (Lines 294-295).

4. In lines 253-254 it is stated "This effect (though not significant) was also observed with the Tegaderm only and polarized groups for the dual-infection wounds, but with MRSA alone". Since the lines above were not clear, this statement is also not clear. Please provide a clearer explanation.

Done.

5. In the results section, the authors state that “the non-polarized group exhibited significantly less purulence than the polarized group in the dual-species infections” (line 262). However, in Figure 5 shows a more negative purulence score change in the polarized group. Is there some data missing? Please clarify

Thank you for this observation; this was our error and has been corrected (Line 303).

6. The results mention “and to a lesser, insignificant amount in the MRSA only infections” (line 263). Where is this shown? Could the authors please explain it better?

This goes hand in hand with the previous comment, and should now be clear.

7. For the statement “Polarized e-bandage treatment produced no observable tissue toxicity” (lines 266-271), is there supporting data? Please provide it.

Supporting data has been added as Table 1.

8. The authors state “Between infected groups, only KC/GRO showed significant elevation in animals treated with polarized vs non-polarized e-bandages” (lines 280-281). However, Figure 6 shows this is only true for the MRSA alone infection. Please clarify.

Clarification has been added (Line 322).

9. In Figure 6 legend it mentions plasma collected from animals treated with and without antibiotics. Where can this difference in treatment be seen in the graph? Please clarify.

Thank you for catching this. No antibiotics were used for these experiments. The figure legend has been corrected.

10. In all the figures, could please the authors include error bars?

Respectfully, we find individual data points to be more informative than error bars. We attempted to add error bars to the figures, as suggested, but they were obscured by data points in some cases (see example below), and added undesirable clutter to others. As such, we left the graph styling as is.

In general, there are some results seen in the graphs that are missing discussion. Concerns about discussion section:

11. In the discussion, lines 313-316 it states, "Between infected groups, animals treated with polarized e-bandages showed significantly elevated levels of KC/GRO when infected with MRSA alone, and insignificantly elevated levels of IL-6 when infected with both MRSA alone and in combination with *P. aeruginosa*." Is this "significantly" or "insignificantly" compared to non-polarized? Please specify.

Clarification has been added (Lines 372-375).

12. Is there previous data on inflammation supporting the findings on the inflammatory response? Could the authors please discuss it?

Explanatory text has been added (Lines 376-382).

13. Although non-polarized bandages do not show significantly improvement compared to Tegaderm only, significant improvement is seen when antibiotics are used (decreased CFUs, increased wound closure, reduced purulence). Could the authors please discuss why non-polarized bandages show some improvement in wound healing when antibiotics are used compared to Tegaderm with antibiotics?

Thank you for making this point. Notably, this was only the case for MRSA infections alone. We have added text to the discussion to address this as much as we can (Lines 363-370).

14. Could the authors please discuss why there is an increase in purulence when antibiotics are added?

The reason for this is unclear. We have addressed this in the Discussion (Lines 386-392).

15. Could the authors comment on the fact that if there is no difference in wound closure or purulence between non-polarized and polarized e-bandages, how do the latter help in wound healing, apart from the reduced bacterial load? (lines 319-321)

Reduction in bacterial load with no host toxicity was the primary goal. However, 48 hours is a short duration for observation of wound healing improvement in mice. Text has been added to the Discussion to address this limitation (Lines 343-353).

Minor concerns:

16. In the Methods and Materials section, it is difficult to visualize the electrochemical bandage and the treatment. Could the authors include an image or scheme to better understand them?

A new figure (Figure 1) has been added. Other figure numbers have been updated accordingly.

17. In the Methods and Materials section, consider moving "Wound biofilm quantification" before "total wound HOCl measurement", to better follow the procedure.

This has been reordered.

18. In line 163, could the authors specify which bacteria is each selective plate for?

This has been added.

19. In line 186, it is unclear how were SEM images scored on bacterial abundance, cell and matrix integrity. Could the authors explain which parameters were used?

Detail has been added (Lines 207-226), and supplemental material provided.

20. Lines 32, 33 and 34: "vs." is missing a point.

This has been corrected.

21. Lines 46-48: This sentence is difficult to understand, please rephrase.

The sentence has been rewritten.

22. Line 50: DNA in the biofilm extracellular matrix is normally referred to as eDNA.

“Extracellular” has been added before the list of EPS components.

23. Line 103: Please be consistent with nomenclature (μL instead of μl)

Standardized to μL .

24. Line 104: “difficult to treat” refers to both PA and MRSA? Then maybe it should say have instead of has.

“Difficult to treat” only refers to the PA isolate. The text has been modified for clarification.

25. Line 140: please state what “IP” refers to

Done.

Re: Spectrum00626-24R1 (HOCl-producing Electrochemical Bandage is Active in Murine Polymicrobial Wound Infection)

Dear Dr. Robin Patel:

I am happy to report that your manuscript has been accepted and that I am forwarding it to the ASM production staff for publication. Your paper will first be checked to make sure all elements meet the technical requirements. ASM staff will contact you if anything needs to be revised before copyediting and production can begin. Otherwise, you will be notified when your proofs are ready to be viewed.

Sincerely,
Cristina Solano
Editor
Microbiology Spectrum